# A Possible Inhibitory Role of Sialic Acid on MUC1 in Peritoneal Dissemination of Clear Cell-Type Ovarian Cancer Cells

**DOI:** 10.3390/molecules26195962

**Published:** 2021-10-01

**Authors:** Yutaka Tamada, Hiroyuki Nomura, Daisuke Aoki, Tatsuro Irimura

**Affiliations:** 1Department of Gynecology, Jyoban Hospital, Tokiwa Foundation, Iwaki 972-8322, Japan; gyokudo2007@yahoo.co.jp; 2Department of Obstetrics and Gynecology, School of Medicine, Keio University, Tokyo 160-8582, Japan; aoki@sc.itc.keio.ac.jp; 3Department of Obstetrics and Gynecology, Fujita Health University, Toyoake 470-1192, Japan; hiroyuki.nomura@fujita-hu.ac.jp; 4Division of Glycobiologics, Intractable Disease Research Center, Graduate School of Medicine, Juntendo University, Tokyo 113-8421, Japan

**Keywords:** mucins, neuraminidase, ovarian neoplasms, peritoneal dissemination, sialic acid

## Abstract

The role of sialic acids on MUC1 in peritoneal dissemination of ovarian cancer cells was investigated. A human ovarian carcinoma cell line, ES-2, was transfected with full-length MUC1 containing 22 or 42 tandem repeats. These transfectants were less adherent to monolayers of patient-derived mesothelial cells than ES-2/mock transfectants. When these cells were inoculated into the abdominal cavity of female nude mice, mice that had received the transfectants showed better survival. When the transfectants were mixed with sialidase and injected, the survival was poorer, whereas when they were mixed with *N*-acetyl-2,3-dehydro-2-deoxyneuraminic acid, a sialidase inhibitor, the survival was significantly prolonged. These behaviors, concerned with peritoneal implantation and dissemination observed in vitro and in vivo, were dependent on the expression of MUC1. Therefore, sialic acid linked to MUC1 in the form, at least in part, of sialyl-T, as shown to be recognized by monoclonal antibody MY.1E12, is responsible for the suppression of adhesion of these cells to mesothelial cells and the suppression of peritoneal implantation and dissemination.

## 1. Introduction

The aberrant expression of mucins and the accompanying glycosylation changes during carcinogenesis and cancer progression have been intensively studied, yet their expression patterns and molecular functions as associated with cancer cell behaviors have not fully been elucidated [1,2]. The association of overexpressed or aberrantly glycosylated Mucin 1 (MUC1) with poor prognosis and malignant behaviors in many types of cancers has been reported [3]. For this reason, MUC1 with unique glycosylation has long been considered as a therapeutic target, and a number of therapeutic chemicals and antibodies have been developed [4,5,6,7,8]. Some of these reagents have been shown to attenuate the putative functions of MUC1. In ovarian cancer progression, the detailed roles and mechanisms of action of MUC1 with unique glycosylation in tumor development and dissemination have yet to be fully understood.

MUC1 is a transmembrane glycoprotein and forms a heterodimer generated from the same gene following self-cleavage in the N-terminal domain of the molecule. MUC1 contains a short intracellular domain at its C-terminal, a transmembrane domain, and a large N-terminal domain bearing repeats of 20-amino acids, which are rich in serines, threonines, and prolines [9,10,11,12]. This extracellular domain is highly glycosylated in a cell- and disease-specific manner, and this complex glycosylation is indispensable for homotypic cellular interactions as well as adhesion and repulsion of MUC1-expressing cells to adjacent tissues [2]. However, the requirements for distinct functions of MUC1 *O*-glycosylation have yet to be systematically studied, due to a lack of common analytical tools. Recent progress in the determination of the requirements for the glycosylation of MUC1 with unique glycans underscores the structure-function relationship of MUC1 glycoforms [13].

We have chosen to study ovarian cancer, which has unique features compared with other types of malignancy. First, cancer cells are highly locomotive during tumor development. Upon peritoneal dissemination, ovarian cancer cells tend to form widespread tumor implants involving the upper abdomen [14]. Such implants are attached to the peritoneal mesothelium, which lines the abdominal cavity, omentum, and bowel serosa, eventually resulting in macroscopic nodules that often cause death due to bowel obstruction. Typically, the tumor remains confined to the abdominal cavity as peritoneal implants as well as free-floating tumor cells contained within ascites, and patients may succumb to the local complications of this disease without ever developing distant metastases. Second, we have previously demonstrated that survival of clear cell-type ovarian cancer cells under anoikis conditions was enhanced by MUC1 expressed on the cell surface [15], thus adhesion may not be necessary for MUC1-positive tumor cells to survive. Third, expression profiles of three sialidases in ovarian cancer cell lines suggest that sialylation of MUC1 oligosaccharides might modulate MUC1-dependent alteration of growth and other behaviors in vitro and in vivo. Finally, the number of tandem repeats serving as glycosylation sites on MUC1 are known to vary from about 30 to 90 as a result of genetic polymorphism [16], and this genetic polymorphism may, in turn, modulate malignant behavior in addition to the many genes associated with the onset and/or development of cancer [17,18].

The present study was thus undertaken to assess the role of sialylation of MUC1 oligosaccharides in peritoneal dissemination of ovarian cancer cells under in vivo and in vitro experimental settings. When ES-2 cells were transfected with MUC1 cDNA, the cells became reactive with monoclonal antibody (mAb) MY.1E12 specific for MUC1 with sialyl-T epitope residing on the GVTSAP sequence on MUC1 tandem repeats, and became less adhesive to human peritoneal mesothelial cells. We also revealed that MUC1 with sialyl-T glycans inhibits peritoneal dissemination of ovarian cancer cells in vivo, as shown in immune compromised mice, uncovering an unexpected role of MUC1 with sialylated *O*-glycans.

## 2. Results

We first established ovarian clear cell carcinoma cell lines overexpressing MUC1 with sialylated glycans by transfecting MUC1 cDNA into ES-2 cells. MUC1 cDNA encoding full-length MUC1 having 22 tandem repeats and 42 tandem repeats were used. Cells exhibiting high cell surface binding of mAb MY.1E12, recognizing sialyl-T epitopes attaching to the threonine residue of the GVTSAP sequence within the tandem repeats [19,20], were selected by repeated cell sorting. Flow cytometric analysis using mAb MY.1E12 confirmed that mock-transfected cells did not express this epitope. Transfectants of MUC1 with 22 tandem repeats (designated as ES-2/T-22 cells) and 42 tandem repeats (designated as ES-2/T-42 cells) were used [15].

One of the main clinical features of ovarian cancer is peritoneal dissemination and implantation. We previously reported that the level of MUC1 with sialyl-T epitopes recognized by mAb MY.1E12 inversely correlated with patient prognosis [15]. We also found that the bindings of mAb MY.1E12 were greater in tumors at advanced clinical stages [15]. To understand the molecular basis for the correlation of MUC1 expression and poor prognosis of ovarian carcinomas, which are often associated with peritoneal dissemination, adhesive behaviors of ES-2 cells to peritoneal mesothelial cells were investigated in the present study. Contrary to our early thoughts, the adhesion of ES-2/T-42 cells and ES-2/T-22 cells was significantly poorer than that of ES-2/mock cells (*p* < 0.05) (Figure 1).

To verify these in vitro findings by in vivo experiments, suspensions of ES-2/mock cells, ES-2/T-22 cells, and ES-2/T-42 cells in 1 mL HBSS were injected into the peritoneal cavity of nude mice. Three weeks after inoculation, mice were sacrificed, and the presence of peritoneal metastasis was examined. Mice that received ES-2/mock cells appeared to experience increased levels of peritoneal dissemination compared to mice that had received ES-2/T-22 cells or ES-2/T-42 cells. Mice that received ES-2/mock cells had small-sized tumors throughout the peritoneal cavity (Figure 2A), whereas mice that received ES-2/T-42 cells had bloody ascites but showed no visible metastatic implantation onto the peritoneal cavity (Table 1, Figure 2B). A significantly smaller tumor burden was found in mice that received ES-2/T-42 cells compared to ES-2/mock cells when the weight of tumor bearing tissues was compared (Figure 2C). These results appeared different from the results shown in our previous report, where cancer cells were subcutaneously injected [15]. The survival rates of mice intraperitoneally inoculated with ES-2/mock, ES-2/T-22 cells, and ES-2/T-42 cells after 3 weeks were 42.9%, 90.9%, and 100%, respectively (Table 1). Intraperitoneal injection of ES-2/T-42 cells expressing MUC1 and high levels of mAb MY.1E12 binding resulted in the production of a large amount of ascites (Table 1). Ascitic growth of ES-2/T-42 cells seemed to be due to increased growth under anoikis conditions, as we previously reported [15].

We hypothesized that reduced adhesion to mesothelial cells and reduced peritoneal dissemination of ES-2/T-22 and ES-2/T-42 cells was due to electrostatic repulsion by sialic acid residues brought about by the expression of MUC1 cDNA. We therefore examined the effect of removal of sialic acids from the surfaces of ES-2/mock cells, ES-2/T-22 cells, and ES-2/T-42 cells on the adhesion of these cells to mesothelial cells in vitro. Sialidase treatment of ES-2/T-22 and ES-2/T-42 cells resulted in a significantly increased number of these cells adhering to mesothelial cells (278.2% and 479.2%, respectively) (Figure 3).

The increase after the sialidase treatment was small when ES-2/mock cells were examined, and the difference was not significant. Similar effects of the removal of sialic acid were observed with three other ovarian cancer cell lines, RMG-I, RMG-II, and KK cells (Appendix A). These three cell lines exhibit various surface levels of MUC1 with sialyl-T epitope (Appendix A). Expression levels of MUC1 with sialyl-T epitope on the cell surfaces of RMG-I, RMG-II, and KK cells seemed to correspond to the degree of the decrease in adhesion.

To assess if sialic acid on ES-2/T-42 cells affects the behavior of these cells in vivo, experiments with nude mice were employed. ES-2/T-42 cells were mixed with sialidase, and mice were intraperitoneally injected with the mixture. ES-2/T-42 cells without the addition of sialidase were used as controls. The survival rate was significantly lower when cells had been injected together with sialidase (*n* = 10) (Figure 4). Therefore, sialic acid residues on ES-2/T-42 cells are likely to act protectively against peritoneal dissemination through preventing peritoneal adhesion and subsequent implantation.

Ovarian clear cell adenocarcinoma cells, including ES-2 cells, were previously shown to express endogenous sialidase [21], which may promote the removal of sialic acid from the cell surface and subsequent peritoneal dissemination. Thus, we tested the effect of injection of *N*-acetyl-2,3-dehydro-2-deoxyneuraminic acid (NADNA), an inhibitor of sialidase. As shown in Figure 4, the survival of mice (*n* = 10) injected intraperitoneally with ES-2/T-42 cells together with NADNA was significantly prolonged compared to mice (*n* = 10) injected with ES-2/T-42 cells alone. The results suggest that this compound may be considered as a drug candidate to prevent peritoneal dissemination of ovarian clear cell carcinomas.

Because the negative charge of sialic acid on MUC1 on ES-2/T-22 cells and ES-2/T-42 cells is responsible for the anti-adhesive properties against mesothelial cells, mesothelial cells apparently express negatively charged groups on their surfaces. Using flow cytometry, we examined the presence of sialylated epitopes on the cell surfaces of mesothelial cells derived from the patient’s omentum. As shown in Figure 5A, the presence of sialyl-T MUC1, revealed by mAb MY.1E12 (Figure 5A4), 2,3-linked terminal sialic acid, revealed by *Maackia amurensis* leucoagglutinin (MAL) (Figure 5A1), and 2,6-linked sialic acid, revealed by *Sambucus sieboldiana* agglutinin (SSA) (Figure 5A2), were detected. The Sialyl-Lewis X epitope, recognized by the binding of mAb KM93, was virtually absent (Figure 5A3). Therefore, it is suggested that sialylated glycans on the surface of ovarian clear cell carcinoma cells prevent adhesion to mesothelial cells through mutual electrostatic repulsion of negatively charged sialic acids.

To prove or disprove such possibilities, the effects of sialidase treatment of mesothelial cells were also assessed. The number of adherent ES-2/T-22 and ES-2/T-42 cells to mesothelial cells significantly increased after sialidase treatment of mesothelial cells (288.4% and 426.5%, respectively), whereas the increase was not significant for ES-2/mock cells (Figure 5B). Sialidase treatment of mesothelial cells also resulted in an increased adhesion of RMG-I and RMG-II cells (273.0% and 266.9%, respectively). The effects on KK cells, which express low levels of MUC1 with sialyl-T epitope, were minimum (106.4%) (Appendix A). These results strongly suggest that sialic acid on MUC1 expressed on the cell surface of ES-2/T-42 and ES-2/T-22 cells is responsible for preventing these cells from adhering to mesothelial cells.

## 3. Discussion

Ovarian malignancies have a worse prognosis than other gynecological malignancies due at least in part to the difficulties of early diagnosis in terms of anatomical location and lack of symptoms at an early stage. However, other causative factors may also be involved, which should be explored to improve the clinical outcomes. Einhorn et al. pointed out that the clinical stage and histologic type are highly important for the prognosis and that the main impact of this malignant disease depends on the peritoneal spreading of tumors [22]. A variety of mucins are known to be expressed in different ovarian carcinomas. As to MUC1, high apical expression was associated with non-mucinous ovarian tumors, and low expression of MUC1 in the apical membrane was reported to be associated with early stage and good outcome for invasive tumors [23]. In our previous study, the most predominant mucin expressed by the ovarian clear cell adenocarcinoma cell lines tested was MUC1 [24]. Moreover, all cell lines derived from clear cell adenocarcinoma expressed cell surface MUC1 with sialyl-T epitopes, recognized by mAb MY.1E12 [24]. Therefore, it is important to assess whether clinical features and biological characteristics of clear cell adenocarcinoma are influenced by the presence of MUC1.

At the advanced stage, ovarian carcinoma cells extend to the surface of the ovarian mass and shed from the primary lesion into the peritoneal cavity. These cells attach to the peritoneal mesothelium that lines the bowel and abdominal wall. It is likely that a variety of adhesion molecules mediate the interaction between an ovarian cancer cell and the peritoneal mesothelium. For example, CD44H, a major receptor for hyaluronic acid on the surface of ovarian cancer cells, was reported to be responsible at least in part for promoting the adhesion of ovarian cancer cells to peritoneal mesothelium [25,26]. However, involvement of other cell surface molecules, such as MUC1, in peritoneal dissemination has yet to be determined. Determination of the role of MUC1 with unique glycans during the process of cancer cell damage to the host is important because MUC1 has recently become an important target of advanced forms of diagnosis and therapy with antibodies and related molecules [4,5,6,7,8].

We developed a method to quantify adhesion of ovarian clear cell carcinoma cells to mesothelial cells and to assess how surface expression of MUC1 with sialylated glycans affects this adhesion. ES-2 cells, which endogenously express low levels of MUC1 on the cell surface [15], were transfected with full-length MUC1 with 22 or 42 tandem repeats. Cells exhibiting high levels of cell surface binding of mAb MY.1E12 were significantly less adherent to mesothelial cell monolayers than ES-2/mock transfectant cells, strongly suggesting that MUC1 with sialoglycans made the cells less adhesive to mesothelial cells (Figure 1). Next, ES-2/mock cells, ES-2/T-22 cells, and ES-2/T-42 cells were inoculated into the peritoneal cavity of nude mice. By monitoring the tumor burden and the survival of mice, we confirmed the inhibitory action of MUC1-bearing sialoglycans against peritoneal dissemination of ovarian cancer cells in vivo. Whether the outcomes observed in this model correspond to the clinical situation has yet to be evaluated, especially because the time courses observed in these mice do not exactly correspond to clinical observations, which may represent one of the limitations of the present study. Further investigations focusing on the essential molecular properties determining the behavior of these cells led us to reveal the importance of sialic acid residues attached to MUC1. Our results showed that the adhesive properties of ES-2/T-42 cells to mesothelial cells increased when the cells were treated with sialidase, and that the survival of nude mice was poorer when ES-2/T-42 cells were injected together with sialidase. Sialidase treatment of three other ovarian cancer cell lines, RMG-I, RMG-II, and KK cells, also resulted in greatly increased adhesive properties in vitro. These findings provide evidence for an inhibitory role of sialic acid residues on MUC1 in peritoneal dissemination of ovarian clear cell adenocarcinoma.

Intraperitoneal injection of ES-2/T-42 cells expressing cell surface MUC1 with sialylated *O*-glycans resulted in the production of a large amount of ascites (Table 1). The accumulation of ascitic fluid did not seem to shorten the survival of the mice. We previously reported that transfection of MUC1 into ES-2 cells induced anoikis resistance in vitro [15]. Recent research showed that the MUC1 extracellular domain confers resistance of epithelial cancer cells to anoikis [27,28]. Our results showing that ascitic growth was seen for ES-2/T-42 cells but not for ES-2/mock cells are in agreement with these observations, and sialylation of the glycans on MUC1 is apparently important in these processes. The molecular basis for ascitic fluid induction remains unknown. It is possible that MUC1 is effective in stimulating vascular permeability of the peritoneum. A similar effect on vascular permeability was also reported for the carbohydrate moiety of the Ebola Zaire virus. The Ebola Zaire virus induced cytotoxic effects in human endothelial cells in vitro and in vivo, mediated through a serine-threonine-rich, mucin-like domain of this type I transmembrane glycoprotein [29].

Thus, sialylation of MUC1 appears to be the key element in reducing peritoneal dissemination and implantation and promoting ascites formation by ovarian carcinoma cells. Sialylation of glycans is known to be involved in antagonizing a variety of adhesive cellular interactions. Wesseling et al. reported that negatively charged sialylated *O*-linked glycans of MUC1 were involved in the inhibition of E-cadherin-mediated cell-cell interactions [30]. From our in vivo findings, we cannot exclude the possibility that sialic acids on molecules other than MUC1 might also be involved in the inhibition of peritoneal dissemination, which can be considered as a limitation of our study. Nevertheless, we have previously reported that ES-2 cells express very low levels of sialylated carbohydrate epitopes, as measured using a panel of carbohydrate-specific monoclonal antibodies [24]. Considering these previous findings, it is likely that on ES-2/T-42 cells, MUC1 represents the main carrier of sialic acid. Experiments aimed at discerning the contribution of individual sialylated molecules on the surface of ovarian cancer cells in peritoneal dissemination should be conducted in the future. Similarly, it remains to be investigated what role MUC1 expressed on the surface of peritoneal mesothelial cells plays in peritoneal dissemination and implantation. Our primary focus in the present paper is on the anti-adhesive effects by repulsion of negative charges carried by sialic acids. Therefore, we did not address possible adhesive effects. There seem to be a variety of adhesive receptors and counter receptors. Previously, we showed that peritoneal mesothelial cells express CD44, ß1-integrin, hyaluronan, and heparan sulfate, but not E-selectin [31]. We also conducted experiments based on the hypothesis that hyaluronan on ovarian cancer cells may bind to CD44 on peritoneal mesothelial cells. As a result, we showed that ovarian cancer cells sorted to have high hyaluronic acid expression exhibited less adhesion to peritoneal mesothelial cells [31]. The involvement of another possible candidate, Siglec-9 [32,33], will be an important subject to be investigated in future studies. Notwithstanding the limitations, the findings of our present study do warrant further examination. In the future, the protective role of MUC1 with sialoglycans in preventing peritoneal dissemination should be further investigated in vivo, employing overexpression, knockdown, or silencing of the MUC1 gene and glycoengineering technologies to express specific MUC1 glycoforms.

In conclusion, the roles of MUC1 with sialyl-T epitopes on ES-2/T42 cells were investigated, and the results showed that sialic acid on MUC1 acts against peritoneal dissemination. The situation resembles the clinical aspects of stage IC ovarian cancer, and our findings may provide important information for the design of new therapeutic strategies for ovarian clear cell adenocarcinoma.

## 4. Materials and Methods

### 4.1. Cells

ES-2 cells derived from a clear cell-type ovarian carcinoma patient [34] were purchased from the American Type Culture Collection (ATCC, Rockville, MD, USA). Cells were cultured in a 1:1 (*v*/*v*) mixture of Dulbecco’s modified Eagle’s minimum essential medium and Ham’s F-12 medium supplemented with 10% fetal calf serum (FCS: Intergen Co., Purchase, NY, USA).

### 4.2. Lectins and Antibodies

Biotinylated MAL, which recognizes sialic acid2-3Galβ1-4GlcNAc, was purified in our laboratory as previously reported [35]. Biotinylated SSA, which recognizes sialic acid2-6Gal/GalNAc, was purchased from Cosmo Bio (Tokyo, Japan). mAb KM93 was purchased from Kyowa Hakko Kogyo (Tokyo, Japan). mAb MY.1E12 [19] was purified from hybridoma cell culture supernatants in our laboratory.

### 4.3. Flow Cytometric Analysis

Flow cytometric analysis was performed using an EPICS XL (Beckman Coulter Inc., Fullerton, CA, USA) to determine the expression of cell surface adhesion molecules. The indirect immunofluorescence method was used to stain cancer cells with primary antibodies or biotinylated lectins (stained for 30 min on ice), followed by the addition of fluorescein-conjugated goat affinity-purified antibody to mouse immunoglobulins (CAPPEL, West Chester, PA, USA) or FITC-Streptavidin (Zymed Laboratories Inc., South San Francisco, CA, USA) as the secondary reagents.

### 4.4. MUC1 cDNA Transfection

A human ovarian carcinoma cell line, ES-2, demonstrated the lowest expression level of MUC1 among all cell lines examined previously [24]; therefore, it was employed to investigate the effect of overexpression of MUC1. MUC1 cDNA encoding full-length MUC1 protein containing 22 or 42 tandem repeats, donated by Dr. O.J. Finn and Dr. M.A. Hollingsworth, respectively [36,37], were ligated into pCEP4 vectors and stably introduced into ES-2 cells using standard electroporation procedures. After selection with hygromycin B (Calbiochem-Novabiochem Co., San Diego, CA, USA), cells exhibiting the highest levels (top 5%) of mAb MY.1E12 binding were obtained by sorting with an EPICS ELITE (Beckman Coulter Inc., Fullerton, CA, USA). After propagation in culture, cell sorting was conducted two more times. ES-2 cells transfected with cDNA encoding MUC1 protein containing 22 or 42 repeats were designated as ES-2/T-22 cells and ES-2/T-42 cells, respectively [15].

### 4.5. Preparation of Peritoneal Mesothelial Cells

Human peritoneal mesothelial cells were obtained from surgical specimens of ovarian carcinoma surgically removed at Keio University Hospital after obtaining informed consent. Mesothelial cells were separated as previously described [31,38,39]. Prior to experimental use, mesothelial cells isolated from different patients were subjected to flowcytometry to confirm the expression levels of CD44, β1-integrin, heparan sulfate, hyaluronan, and E-selectin [31].

### 4.6. In Vitro Peritoneal Implantation Model

The adhesion of ovarian cancer cells to monolayers of mesothelial cells was performed as described previously [31].

### 4.7. In Vivo Peritoneal Dissemination Model

Female KSN nude mice (5 weeks old) were purchased from Japan SLC Inc. (Hamamatsu, Japan) and kept under specific-pathogen-free conditions. Viable ES-2/mock, ES-2/T-22, and ES-2/T-42 cells (2 × 10^6^ cells) in 1 mL Hank’s balanced salt solution (HBSS; Nissui Pharmaceutical Co., LTD., Tokyo, Japan) were injected into the abdominal cavity of mice. Three weeks after the inoculation, mice were sacrificed and examined for the presence of peritoneal dissemination. As an evaluation method that is conscious of actual clinical surgery, the mesentery, omentum, peritoneum, ovary, and uterus of the mice were resected together with implanted tumors, and their cumulative weights were measured.

### 4.8. Sialidase Treatment

Detached cancer cells were treated with 0.5 unit/mL sialidase from *Clostridium perfringens* (Sigma Aldrich, St. Louis, MO, USA) in DMEM/Ham’s F12 containing 10 mM HEPES pH 7.2 at a density of 5 × 10^6^ cells/mL for 30 min at 37 °C, followed by BCECF-AM labeling for 30 min at 37 °C. As described above, labeled cancer cells were used for in vitro adhesion assays. Removal of sialic acid residues from MUC1 and other cell surface molecules was confirmed by flow cytometry using mAb MY.1E12 and biotinylated MAL and SSA before and after sialidase treatment of cells.

### 4.9. In Vivo Peritoneal Dissemination Model to Evaluate the Effect of Sialidase or NADNA on the Overall Survival of BALB/c-nu/nu Mice

BALB/c-*nu/nu* female mice (5 weeks old) were purchased from CLEA Japan Inc. (Tokyo, Japan) and kept under specific-pathogen-free conditions. Viable ES-2/T-42 cells (2 × 10^6^ cells) in 1 mL PBS with or without sialidase (2 units) were injected into the abdominal cavity of BALB/c-*nu/nu* mice. In another experiment, viable ES-2/T-42 cells (2 × 10^6^ cells) in 1 mL PBS with or without NADNA (100 μM) (Sigma-Aldrich Japan, Tokyo, Japan) were injected into the abdominal cavity of BALB/c-*nu/nu* mice.

### 4.10. Statistical Analysis

Differences in adhesion of ES-2/mock cells, ES-2/T-22 cells, and ES-2/T-42 cells to peritoneal mesothelial cells were analyzed using the Kruskal-Wallis rank sum test. Differences in tumor-bearing tissue weight were analyzed using the unpaired Student’s *t*-test after confirming that the data followed a normal distribution. Differences in cell-cell adhesion in vitro after sialidase treatment were analyzed using the unpaired Mann-Whitney U-test. Differences in three week survival rates and ascites production rates of nude mice were statistically tested using Fisher’s exact probability test. Significant differences between Kaplan-Meier survival curves of nude mice were tested using the log-rank test. A *p*-value of less than 0.05 was considered statistically significant.

## Figures and Tables

**Figure 1 molecules-26-05962-f001:**
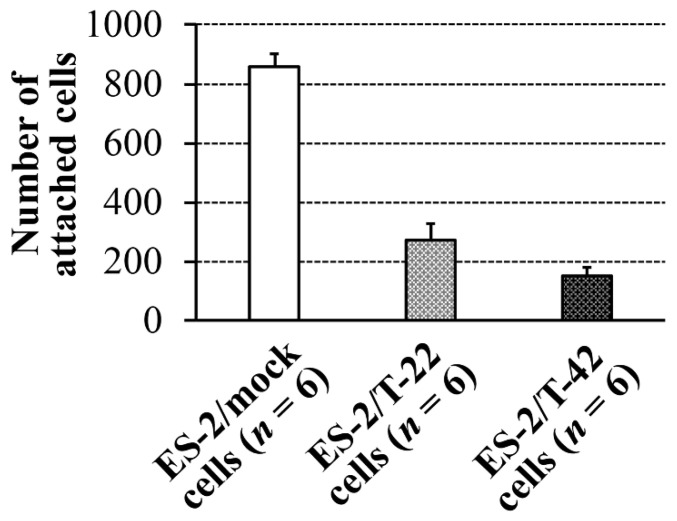
Adhesion of ES-2/mock cells, ES-2/T-22 cells, and ES-2/T-42 cells to peritoneal mesothelial cells. The *Y*-axis indicates the number of cells attached to the mesothelial cell monolayer 30 min after seeding. Data shown are mean ± SEM. Kruskal-Wallis rank sum test. *p* < 0.05.

**Figure 2 molecules-26-05962-f002:**
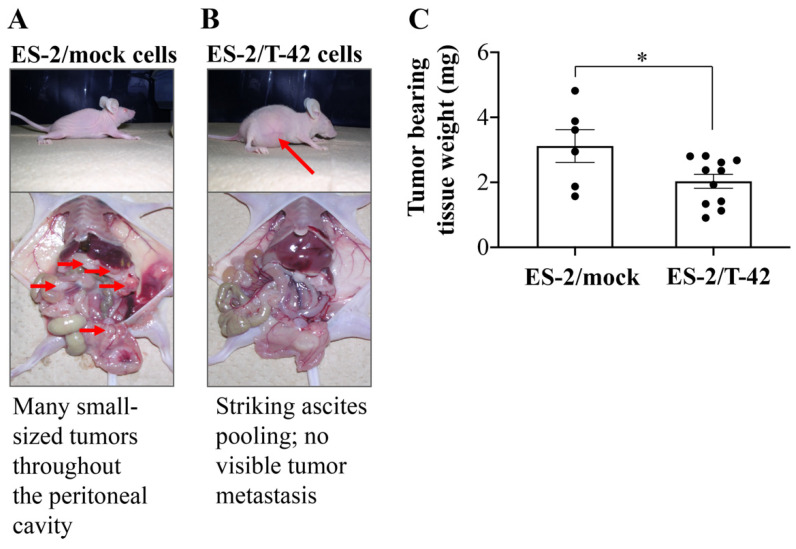
Patterns of dissemination of ES-2/mock cells and ES-2/T-42 cells in vivo. Nude mice were inoculated with ES-2/mock cells, ES-2/T-22 cells, or ES-2/T-42 cells. (**A**) Macroscopic and intraperitoneal findings of nude mice inoculated with ES-2/mock cells. Small red arrows indicate the position of tumors. (**B**) Macroscopic and intraperitoneal findings of nude mice inoculated with ES-2/T-42 cells. The big red arrow indicates a bloated abdomen due to accumulated ascites. (**C**) Tumor burden in mice intraperitoneally injected with ES-2/mock or ES-2/T-42 cells three weeks after cell inoculation. Mesentery, omentum, peritoneum, ovary, and uterus of the mice were resected together with implanted tumors, and their cumulative weights were measured. Each dot represents one mouse, and mean ± SEM are shown. Unpaired Student’s *t*-test. * *p* < 0.05.

**Figure 3 molecules-26-05962-f003:**
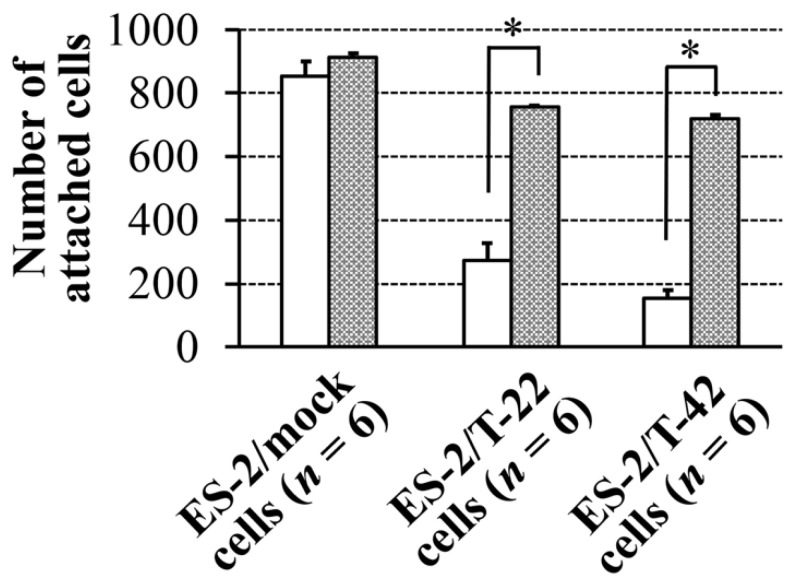
Effect of sialidase treatment of ES-2/mock, ES-2/T-22, and ES-2/T-42 cells on their adhesion to mesothelial cells. Data shown are mean ± SEM. Mann-Whitney U-test. * *p* < 0.05.

**Figure 4 molecules-26-05962-f004:**
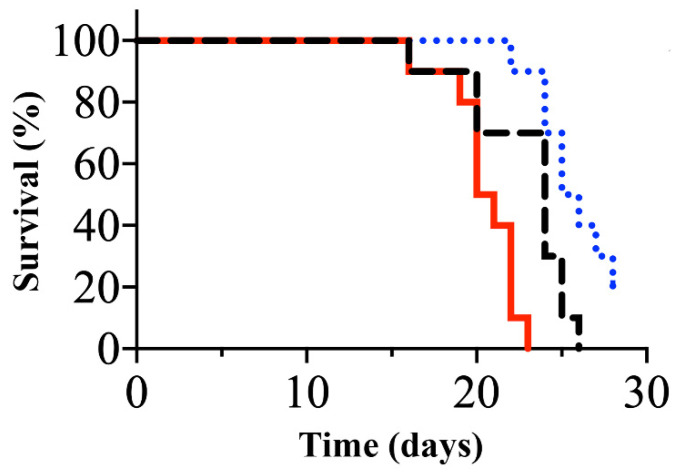
Kaplan-Meier survival curves of nude mice after abdominal inoculation with ES-2/T-42 cells alone (black dashed line), ES-2/T-42 cells together with sialidase (red solid line), and ES-2/T-42 cells together with *N*-acetyl-2,3-dehydro-2-deoxyneuraminic acid (NADNA), a sialidase inhibitor, (dotted blue line). Ten mice were used in each group. Kaplan-Meier method log rank (Mantel-Cox) test. *p* < 0.0001.

**Figure 5 molecules-26-05962-f005:**
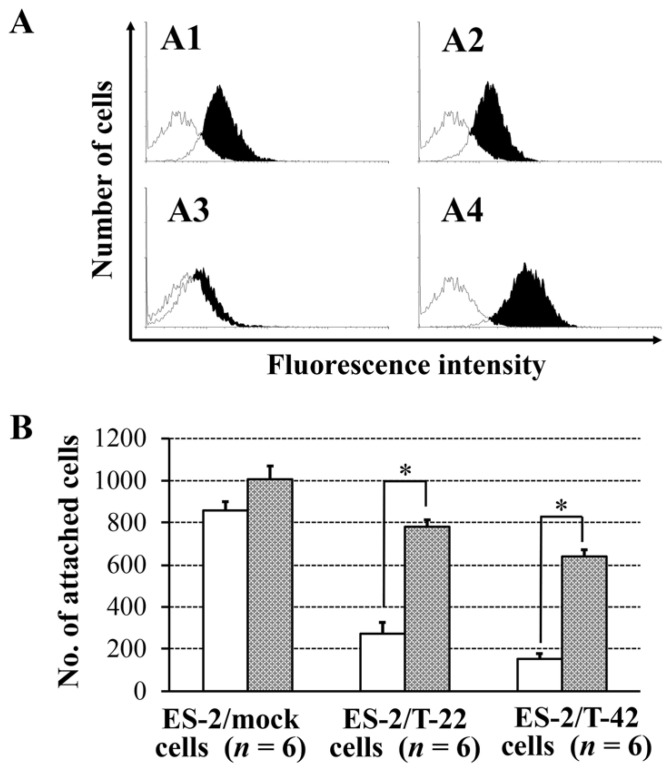
Effect of sialic acids expressed on the surface of peritoneal mesothelial cells on the binding of MUC1 transfected ovarian cancer cells. (**A**) Bindings of lectins specific for sialylated carbohydrate chains and antibodies specific for sialylated epitopes of MUC1 to the cell surfaces of peritoneal mesothelial cells derived from ovarian cancer patients. (**A1**) *Maackia amurensis* leucoagglutinin (specific for α2,3-linked sialic acid), (**A2**) *Sambucus sieboldiana* agglutinin (specific for α2,6-linked sialic acid), (**A3**) mAb KM93 (specific for sialyl Lewis X), and (**A4**) mAb MY.1E12 (specific for sialyl-T MUC1). Flow cytometric analysis. No shading: negative control (secondary antibody only); black shading: antibody/lectin bound cells. (**B**) Effect of sialidase treatment of mesothelial cells on the adhesion of ES-2/mock, ES-2/T-22, and ES-2/T-42 cells to the peritoneal mesothelial cell layer. Data shown are mean ± SEM. Mann-Whitney U-test. * *p* < 0.05.

**Table 1 molecules-26-05962-t001:** Three-week survival rate and incidence of ascites in in vivo peritoneal dissemination model assay ^a^.

	ES-2/*Mock* (*n* = 14)	ES-2/T-22 (*n* = 11)	ES-2/T-42 (*n* = 11)
Three-week survival rate (%)	42.9	90.9 *^1^	100 *^1^
Incidence of ascites (%)	0 *^2^	18.2 *^2^	81.8

^a^ Viable ES-2/mock, ES-2/T-22, and ES-2/T-42 cells (2 × 10^6^ cells) in 1 mL Hank’s balanced salt solution were injected into the abdominal cavity of KSN nude mice (5 weeks old). Fisher’s exact probability test. * *p* < 0.05. *^1^ indicates a significant difference compared to ES-2/mock; *^2^ indicates a significant difference compared to ES-2/T-42.

## Data Availability

Data is contained within the article and its Appendix A.

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
