# Peer review of "A Possible Inhibitory Role of Sialic Acid on MUC1 in Peritoneal Dissemination of Clear Cell-Type Ovarian Cancer Cells"

_molecules, 2021, doi:10.3390/molecules26195962_

Round 1

Reviewer 1 Report

The paper entitled “A Possible Inhibitory Role of MUC1 with Sialyl-T epitopes in Peritoneal Dissemination of Clear Cell-type Ovarian Cancer Cells” by Tamada, Y. et al. investigates the role of MUC1 sialylation degree in the spread of ovarian cancer cells both in vitro and in vivo. The authors demonstrate that the lack of sialylation in MUC1 expressing cancer cells is correlated with the cancer cell spreading to peritoneal adhesions. Furthermore, the role of exogenous sialidase in cancer cells was assessed and the use of sialidase inhibitors decreased the spreading of tumour cells.

The experiments are well performed and described. The conclusions obtained are meaningful in the context of understanding the role of glycosylation of MUC1 in cancer progression. Nevertheless, some points need to be addressed before publication.

Major point.

The purpose of the experiment of FC measurements of mesothelial cells derived from patient’s omentum is not clear to me. In the experiment, the different expression of sialic acids is efficiently studied along with the lack of expression of Sialyl Lewis X by FC. I don´t see how this expression would correlate with the cell spreading. It would be more interesting to study the expression of receptors that could be involved in the recognition of Gal/GalNac epitopes (Galectins???) that would be exposed after sialic acid removal and how they could contribute to the cell- cell communication between cancer and mesothelial cells. At least the experiment should be put more into context discussing about the significance of it.

Minor points.

After the sentence “For this reason, MUC1 has long been considered as a therapeutic target and a number of therapeutic chemicals and antibodies have been developed” references of the mentioned previous reports should be included.

The following sentences are confusing, a clearer explanation of previously reported results should be included.

We previously reported that the level of MUC1 with sialyl-T epitopes recognized by mAb MY.1E12 inversely correlated with patient prognosis [10]. However, the antibody binding was also associated with the clinical stage of ovarian cancer [10]

In the experimental section, the details for the isolation of Maackia Amurensis leukoagglutinin (MAL) or a reference of the procedure employed should be included.

Reviewer 2 Report

The main objective of the present paper was to investigate the effects of the expression and glycosylation of the glycoprotein mucin 1 (MUC1) by breast cancer cells, on their adhesion to mesothelial cells. A cell line derived from ovarian carcinoma – ES-2 – was chosen. These cells express low levels of MUC1 and were transfected with MUC1 cDNAs encoding full-length MUC1 having 22 tandem repeats and 42 tandem repeats. The focus of this paper was the presence of sialyl-T epitopes, which are recognized by mAb MY.1E12, and inversely correlated with patient prognosis. Both experiments in vitro and in vivo were performed, and the authors concluded that MUC1 with sialyl-T epitopes acts against peritoneal dissemination.

Nevertheless, it seems that the literature citation is a little bit outdated: only one paper published in 2019 was cited, and 20 (out of the 30) of the cited papers were published more than 20 years ago.

Particularly, mechanisms were not proposed, and not discussed. For instance, there are important papers on MUC1 glycosylation and action in cancer that were completed disregarded by the authors. Important papers were published in 2014 and 2017 (Zhao, Q., Piyush, T., Chen, C. et al. MUC1 extracellular domain confers resistance of epithelial cancer cells to anoikis. Cell Death Dis 5, e1438 (2014). https://doi.org/10.1038/cddis.2014.421; Yu, LG. Cancer cell resistance to anoikis: MUC1 glycosylation comes to play. Cell Death Dis 8, e2962 (2017). https://doi.org/10.1038/cddis.2017.363) showing that MUC1 plays an important role in epithelial cancer cell resistance to anoikis. Resistance to anoikis is a hallmark of oncogenic epithelial–mesenchymal transition and contributes prominently to tumorigenesis and to metastasis by allowing survival of cancer cells in the blood or lymphatic circulation and thus facilitating their metastatic spread to remote sites. Does the sialyl-T epitopes are important to induce/prevent this effect? This is a relevant point, that should have been discussed in the present paper.

Reviewer 3 Report

The authors showed that sialy-T antigen expressed on the MUC1 plays important role in enhancing the peritoneal dissemination of ovarian cancer cells. The role of sialyl-Tn antigen on the metastasis of cancer was well-known by previous studies. However, the importance of sialyl-T antigen in metastatic ovarian cancer was not much studied. Thus, the study has novelty and merit in the field of cancer glycobiology. However, the manuscript should be improved on several points.

Major points.

1. In Figure 3, the FACS analysis using several lectins and Abs could not specify that the sialyl-T antigen is the most prominent change on the MUC1 glycosylation. To specify the major epitope, the authors should check more sialylated epitopes, such as sialyl-Tn, sialyl core1, and sialyl Lewis antigens.

2. If the sialy-T antigen is a major epitope on the MUC1, the treatment of sialidase is not sufficient to support the authors' claim that sialyl-T antigen plays a key role in the dissemination of ovarian cancer. To clarify the claim, the authors should specify which genes are involved in making sialyl-T antigen and knock down the genes. In the synthesis of sialyl-T antigen, ST6GalNAcs and ST3Gal1/2 enzymes are involved. 

3. In Figure 5, the dashed line in the A and B panels are equal. Please collect the panel A and B in one plot. In addition, the dissemination of ovarian cells should be shown in Figure 5, as like in Figure 2 and Table 1. 

Minor point.

4. Which is the receptor for ST antigen on the mesothelium? In addition to E-selectin, a receptor for sialyl Lewis antigens, the previous studies showed that Siglec-9 is expressed in the human mesothelial cells. (PMID 30542051 and 20497550) The issue should be included in the discussion section.

5. In Figure 2B, the results should be presented in Kaplan-Meier survival plots.

Round 2

Reviewer 1 Report

The revision has fulfilled all my concerns about the manuscript and therefore, I suggest the acceptance of the revised manuscript.

Reviewer 3 Report

The questions raised by the reviewer were properly solved. The manuscript was also well revised. I recommend the publication of this manuscript.